# Massive Thrombosis of Mitral Bioprosthesis Due to SARS-CoV-2 Infection

**DOI:** 10.3390/jcm11185277

**Published:** 2022-09-07

**Authors:** Mariateresa Librera, Stefania Paolillo, Guido Carlomagno, Gianluca Santise, Antonio Mariniello, Saverio Nardella, Carlo Briguori, Daniele Maselli

**Affiliations:** 1Cardiac Multi-Imaging Unit, Mediterranea Cardiocentro, Via Orazio 2, 80122 Naples, Italy; 2Department of Advanced Biomedical Sciences, Federico II University of Naples, Via Pansini 5, 80131 Naples, Italy

**Keywords:** heart valve prosthesis, thrombosis, COVID-19, SARS-CoV-2

## Abstract

Thromboembolic events have been reported as frequent and fearsome complications in patients affected by SARS-CoV-2 infection. Patients undergoing cardiac valve replacement exhibit an increased risk of valve thrombosis, even with prosthetic biological valves, and especially in the first period after surgery. The management of these patients is challenging and requires prompt interventions. We report the case of a young woman infected by SARS-CoV-2 three months after double cardiac valve replacement that developed a massive prosthetic biological valve thrombosis despite optimal anticoagulant therapy.

## 1. Introduction

SARS-CoV-2 infection has been related to the occurrence of thromboembolic events with a multifactorial etiology [1]. Patients undergoing cardiac valve replacement are exposed to an increased risk of valve thrombosis, even with biological tissue valves, especially early after surgery [2]. We describe the clinical case of a young patient in whom SARS-CoV-2 infection three months after double valve replacement led to the development of massive prosthetic thrombosis despite optimal anticoagulation.

## 2. Case Presentation

We report the case of a 53-year-old woman, affected by diabetes mellitus, hypertriglyceridemia, and hypothyroidism, with a previous history of Hodgkin lymphoma treated with chemotherapy, paroxysmal atrial fibrillation complicated by an ischemic stroke without residual effects, and subsequent evidence of patent foramen ovale (PFO). On 11 August 2020, the patient underwent cardiac surgery at another institution, with aortic valve replacement (Edwards Perimount Magna Ease 21) for severe aortic stenosis, mitral valve replacement for severe mitral regurgitation (Edwards Perimount Magna Ease 27), coronary artery bypass graft for a critical right coronary artery stenosis (saphenous vein graft to right coronary artery), and direct closure of PFO. The preoperative transthoracic echocardiogram (TTE) showed biventricular dysfunction (left ventricular ejection fraction—LVEF 30%) and a tricuspid annular excursion (TAPSE) of 12 mm. The surgical procedure was uneventful, without significant complications; the patient was moved to a cardiac rehabilitation center on the 9th postoperative day and was discharged home on the 20th postoperative day, on oral anticoagulation with warfarin (therapeutic range 2.5–3.5). The pre-discharge TTE showed an LVEF of 40%, reduced TAPSE, compatible with the postoperative phase, and mean gradients of 7 mmHg on the mitral prosthesis (due to a prosthesis/patient mismatch) and 13 mmHg on the aortic prosthesis. Three months later, at a routine outpatient visit, INR was in the therapeutic range (2.9), and the TTE showed improved LV systolic function (EF 52%), normal right ventricular (RV) function (TAPSE 23 mm), a normal gradient on the aortic prosthesis, and an increase in mitral prosthesis gradient (12 mmHg vs. 7 mmHg on discharge). Two weeks thereafter, on17 December 2021, the patient was admitted to the Emergency Room for dyspnea and was diagnosed with bilateral SARS-CoV-2-related interstitial pneumonia with moderate pleural effusion. She was treated with corticosteroids with a progressive improvement of symptoms and clinical status, discharged after about 20 days (8 January 2021) with a nasopharyngeal swab still positive for SARS-CoV-2 infection, and entrusted to the SARS-CoV2 domiciliary care services. Oral anticoagulant therapy with warfarin was continued during the hospital stay and after discharge. On 22 January 2021, the patient was newly admitted at the Internal Medicine Department for dyspnea; the cardiological consult and the TTE revealed the presence of mitral prosthesis dysfunction (mean gradient 18 mmHg) with a suspected thrombotic apposition. Thus, a transesophageal examination (TEE) was performed that confirmed the presence of a thrombosis of the ventricular mitral face, with a severe reduction of the valvular area (0.9 cm^2^), a severe increase of mean gradient, and a diffuse thrombosis of the left atrium, excluding the appendage. INR in this case was also in the therapeutic range (value 3.0 with 12.5 mg of warfarin/week, goal 2.5–3.5); a timeline graph of events and INR levels is reported in Figure 1.

On 19 February 2021, the patient was referred to our center for a cardio-surgical assessment. At admission, the patient was in stable hemodynamic status andsymptomatic for dyspnea on minimal effort; however, she exhibited no orthopnea or resting dyspnea. The clinical assessment revealed the presence of diffuse ecchymosis, ischemia of the third toe of the right lower limb, and pulmonary basal rales; the EKG showed sinus tachycardia. Lab analysis showed an INR value of 3.5, normal renal function, increased white blood count with prevalence of neutrophils (14.280/μL, 82%), thrombocytopenia (50,000/μL), and 9-fold increase of D-Dimer (4.31 mg/L). A total-body CT scan excluded pulmonary embolism, revealed the presence of lung interstitial thickening and mild bilateral pleural effusion, and showed eccentric occlusive bilateral thrombosis of femoral veins, more prevalent on the right side; the cardiac phase showed extensive thrombotic stratifications in the left atrium (Figure 2). The TTE and TEE (Figure 3) showed mildly reduced systolic function (EF 50%) and gross and diffuse thrombosis of mitral prosthetic leaflets, prevalently on the ventricular side (thickness of approximately 1 cm), resulting in leaflet restriction and severe stenosis (mean gradient of 15 mmHg) with a severely reduced orifice area using 3D-guided planimetry (0.8 cm^2^). Moreover, a stratified thrombus (thickness of approximately 2 cm) was diffusely visible on the left atrial free wall. The aortic prosthesis had a mean gradient of 15 mmHg in the absence of thrombosis. The estimated systolic pulmonary artery pressure was 38 mmHg; an acceptable RV function (S’ 11 cm/s) was measured.

The coronary angiogram showed a patent venous bypass, and on 23 February 2021, the patient underwent redo mitral valve replacement (Mosaic n.27) without immediate perioperative complications; the day after, she was extubated and the TTE confirmed the presence of an euvolemic status without prosthetic complications, mild LV systolic dysfunction, and reduced RV function, compatible with the postoperative status. After a few hours, the patient exhibited a contraction of diuresis not responsive to high-dose i.v. diuretics or enoximone; thus, continuous veno-venous haemofiltration (CVVH) was initiated, along with inotrope support (noradrenaline and enoximone). Repeat TTEs showed a severe dilation of the right chambers, with severe RV dysfunction, severe tricuspid regurgitation with an estimated sPAP of 65 mmHg, paradoxical diastolic interventricular septum motion, compatible with a status of RV pressure and volume overload, inferior venous cava dilation without respiratory excursion, and a collapsed LV cavity. The lung CT scan excluded the presence of pulmonary embolism. At this time, extracorporeal membrane oxygenation (ECMO) was started. During the night, the patient, who was still anuric and hypotensive (mean arterial pressure 65 mmHg) with progressive increase in lactates, was newly intubated and mechanically ventilated. In the next few hours, a progressive decline in clinical conditions was observed, with the occurrence of pulmonary edema, mydriasis, extreme bradycardia, and hypotension. On 25 February 2021, the patient died.

## 3. Discussion

We describe a rare case of a patient with a recent SARS-CoV-2 infection presenting with severe thrombosis involving the mitral biological prosthesis and the left atrium.

Thrombotic complications such as venous thromboembolism, deep vein thrombosis, pulmonary embolism, or right ventricular thrombosis are common in patients with SARS-CoV-2 infection, while arterial thromboembolism is less frequently described, as well as thrombosis of biological prostheses [1,2,3,4]. When our patient was diagnosed with bioprosthesis and left atrium thrombosis, she presented with a sinus rhythm and was still being treated with warfarin, with INR in the therapeutic range [5]. This was in accordance with SARS-CoV-2 infection treatment recommendations because she was still in the first three months after surgical valve replacement. Risk factors for prosthetic valve thrombosis were left atrial dilatation and a state of hypercoagulability, demonstrated by high D-Dimer and fibrinogen levels, probably due to the previous SARS-CoV-2 infection. Current knowledge regarding SARS-CoV-2-related disease suggests that the initial events occur in the lung; thus, a severe inflammatory response originating in the alveoli triggers a dysfunctional cascade of inflammatory thrombosis in the pulmonary vasculature, leading to a state of local coagulopathy followed by a generalized hypercoagulable state, which results in macro- and microvascular thrombosis. Lab analysis of our patient also showed severe thrombocytopenia, a well-recognized as a marker of worse outcomes; thrombocytopenia is common in patients with severe SARS-CoV-2 infection, and it is unclear whether a decreased platelet count reflects a more severe disseminated coagulopathy with increased consumption or a direct viral infection of the platelets [6].

After undergoing redo mitral valve replacement (Mosaic n.27), the patient presented RV dysfunction with cardiogenic shock. It was not due to pulmonary embolism, as demonstrated by the lung CT scan, and, despite the optimization of preload and of contractility with vasopressors and inotropes, she needed an ECMO support in order to provide indirect help to the RV by reducing wall tension, delivering oxygenated blood to the coronary circulation, and letting the lung rest.

Apart from the clinical interest, the present case raises important issues regarding the management of SARS-CoV-2 infection in small and suburban health care centers that have no access to specialistic consultation or advanced imaging technique examinations. Our patient was not evaluated for SARS-CoV-2 infection by a cardiologist during the hospital stay, probably delaying the diagnosis of prosthetic valve dysfunction that, if treated earlier, might have saved her life. SARS-CoV-2-infected cardiac or cardio-surgical patients need to be prematurely referred to specialized centers.

## 4. Conclusions

SARS-CoV-2 infection may lead to severe thrombotic complications in patients who have recently undergone cardiac valve surgery, considerably complicating the therapeutic management. Cardio-surgical patients infected by SARS-CoV-2 need prompt and continual cardiological consultations in order to prematurely identify the occurrence of cardiac complications and need to be referred as soon as possible to specialized centers.

## Figures and Tables

**Figure 1 jcm-11-05277-f001:**
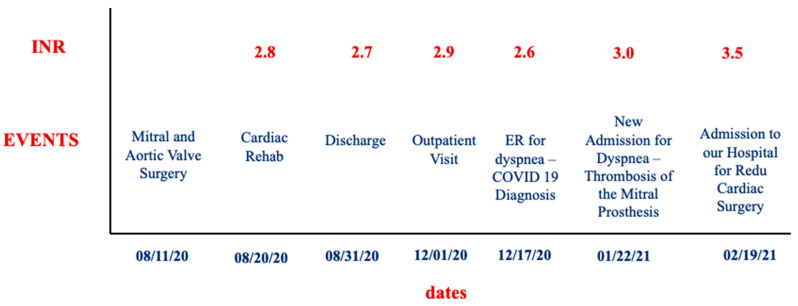
Timeline graph of events and INR levels. ER: emergency room.

**Figure 2 jcm-11-05277-f002:**
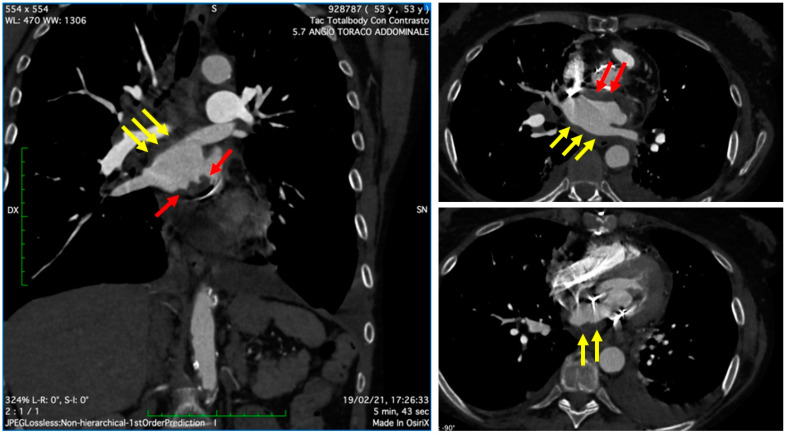
Cardiac CT showing extensive thrombotic stratifications in the left atrium, both on the posterior free wall (*yellow arrows*) as well as in the peri-annular region near the prosthesis ring (*red arrows*).

**Figure 3 jcm-11-05277-f003:**
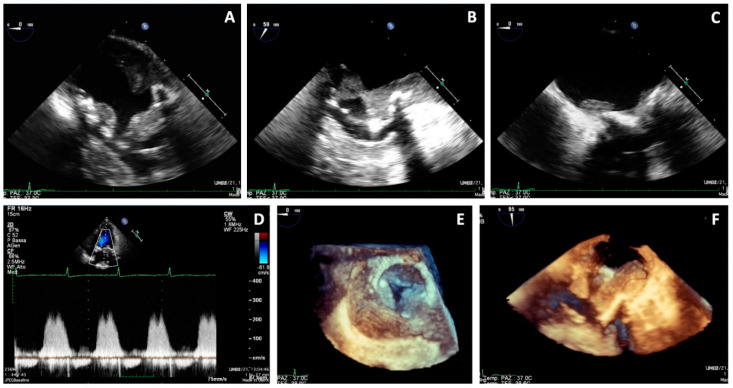
Transesophageal echocardiography showing massive thrombosis of mitral prosthetic leaflets, prevalently on the ventricular side, a stratified thrombus on the left atrial free wall (**A**–**C**,**F**), increased gradients on the mitral prosthesis (**D**), and reduced orifice area at 3D en-face view (**E**).

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
