# Peer review of "Massive Thrombosis of Mitral Bioprosthesis Due to SARS-CoV-2 Infection"

_jcm, 2022, doi:10.3390/jcm11185277_

Round 1

Reviewer 1 Report

Librera et al. present a case of a young women infected by SARS-CoV-2 three months after double 20 cardiac valve replacement that developed a massive prosthetic biological valve thrombosis despite 21 optimal anticoagulant therapy. Thrombosis is known to be more prevalent in COVID-19 patients. However, this case does a nice job highlighting how it can impact people with bioprosthetic valves. Overall well written, may comments below.

Introduction:

-Well written.

Case:

-Please shorten the case description. Superfluous details can be removed, such as laboratory date or TTE data. Should be overall concise.

-Was there any consideration of treating the patient empirically with IV heparin or tpa infusion before undergoing for repeat surgery.

Discussion

-Is there any data that DOAC may be better than warfarin for COVID-19 patients? If so, please discuss.

Figures

-There is no key/legend for Figure 1.

Author Response

We thank the Reviewer for the interest to our manuscript and for the useful comments that significantly helped to improve paper quality.

Please find below a point by point response to you comments.

1. Please shorten the case description. Superfluous details can be removed, such as laboratory date or TTE data. Should be overall concise.

We thank for the suggestion and now shortened the test as suggested, especially removing some superfluous TTE data.  

2. Was there any consideration of treating the patient empirically with IV heparin or tpa infusion before undergoing for repeat surgery.

We thank the Reviewer for this important point. Our medical and surgical team considered to empirically treat the patient with IV heparin, however because of the extensive and massive thrombosis, and the severe clinical and laboratory patient's state, we finally decided to directly perform redo mitral valve replacement.

3. Is there any data that DOAC may be better than warfarin for COVID-19 patients? If so, please discuss.

Regarding this specific point, no data to support the use of DOAC in COVID-19 thrombosis ar at the moment available. Recently, the ISTH DIC subcommittee published a communication on anticoagulation in COVID-19 (J Thromb Haemost. 2020; 18:2138–2144) and there was no mentioning of the use of DOACs. Thus, the use of DOACs in hospitalized COVID-19 patients is not straightforward and no specific data are available for patients with thrombosis of valve prosthesis (Hemasphere. 2021 Jan; 5(1): e526). 

4. There is no key/legend for Figure 1.

Now included.

Reviewer 2 Report

It is a very interestting case and the case has been written very well. i only have few minor issues:

1. Use abbreviation only once in the beginning  for example TTE full abbreviation is use multiple times

-          2. Author mention “Three months later, 50 at a routine outpatient visit, INR was in the therapeutic range” , what was goal and how much warfarin?

-        3.   Please write date in a proper format consistent throughout the manuscript. E.g December 25, 2021

-          4. What was INR levels throughout the course of hospital stay. I think a timeline graph of events and INR levels will be very interesting to the readers.

Author Response

We thank the Reviewer for the interest to our manuscript and for the useful comments that significantly improved the quality of the paper.

Please find below a point by point response to you comments.

1. Use abbreviation only once in the beginning  for example TTE full abbreviation is use multiple times

We thank for the observation and now corrected as suggested.

2. Author mention “Three months later, 50 at a routine outpatient visit, INR was in the therapeutic range” , what was goal and how much warfarin?

These data have been included in the revised manuscript. 

3. Please write date in a proper format consistent throughout the manuscript. E.g December 25, 2021

Corrected as suggested

4. What was INR levels throughout the course of hospital stay. I think a timeline graph of events and INR levels will be very interesting to the readers.

Thank you for the suggestion. We now included a timeline graph of INR values and events (new Figure 1)
